# Hepatic Sinusoidal Obstruction Syndrome Secondary to Aplastic Anemia/Paroxysmal Nocturnal Hemoglobinuria Syndrome: A Rare Case

**DOI:** 10.3390/diagnostics15212712

**Published:** 2025-10-27

**Authors:** Xinyue Liu, Xiaohe Li, Yanhua Chen, Rui Huang

**Affiliations:** 1Department of Nephrology, Peking University People’s Hospital, Beijing 100044, China; lxysdjnyz@163.com; 2Peking University Hepatology Institute, Peking University People’s Hospital, Beijing 100044, China; lxhe0903@163.com; 3School of Basic Medical Sciences, Peking University Health Science Center, Beijing 100191, China; cyaoh66@163.com

**Keywords:** hepatic sinusoidal obstruction syndrome, aplastic anemia, paroxysmal nocturnal hemoglobinuria, CT

## Abstract

Paroxysmal nocturnal hemoglobinuria (PNH) is associated with bone marrow failure disorders and may arise during the long-term follow-up of aplastic anemia (AA), which is named AA/PNH syndrome. Thrombosis is the most frequent clinical complication and is the main cause of mortality in PNH. However, thromboses tend to originate in hepatic and cerebral venous vessels, but rarely in the hepatic microvascular vein in PNH patients. Here, we report on a young man with hepatic sinusoidal obstruction syndrome (HSOS) secondary to AA/PNH syndrome. His main manifestations were hemolytic anemia, renal injury, ascites, hepatomegaly, and elevated liver enzymes. The diagnosis was confirmed by peripheral blood flow cytometry, enhanced computed tomography (CT), and liver biopsy. Initially, he received symptomatic treatments including diuretics, intermittent abdominal paracentesis, and low-molecular-weight heparin. Meanwhile, due to the occurrence of PNH activity during hospitalization, methylprednisolone 40 mg per day was administered, which was later transitioned to oral prednisolone. Subsequently, the dose of corticosteroids was gradually decreased once his hemoglobin stabilized. The association between HSOS and AA/PNH syndrome is exceptionally rare, as evidenced by the scant literature on the subject. This case underscores the critical need for awareness of HSOS secondary to AA/PNH syndrome, which needs a high index of suspicion and for which prompt treatment is needed to reduce morbidity and mortality.

**Figure 1 diagnostics-15-02712-f001:**
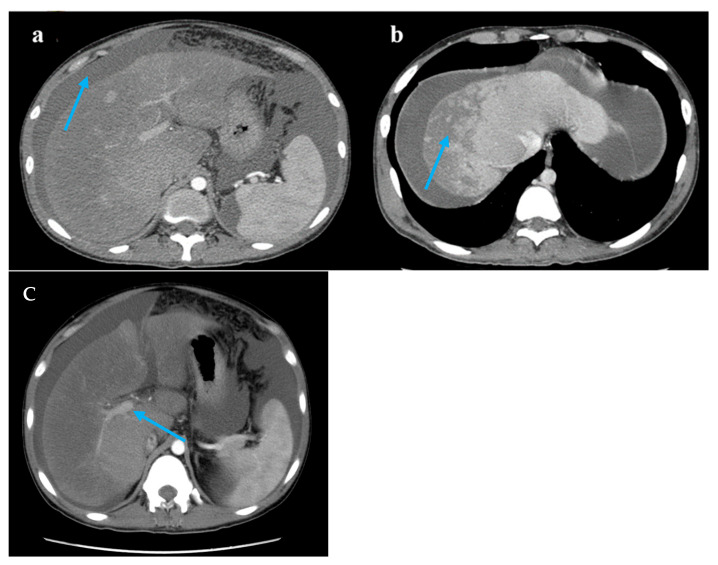
Computed tomography images. Legend: (**a**) A large accumulation of fluid in the abdomen and pelvis (blue arrow). (**b**) The liver lobes exhibit balanced proportions. Diffuse, non-uniform reduction in liver density is observed. Areas of reduced density show low contrast enhancement on contrast-enhanced scans. Multiple nodules are visible within the low-density regions (blue arrow). (**c**) The hepatic veins are thin, and the portal vein is less full (blue arrow).

**Figure 2 diagnostics-15-02712-f002:**
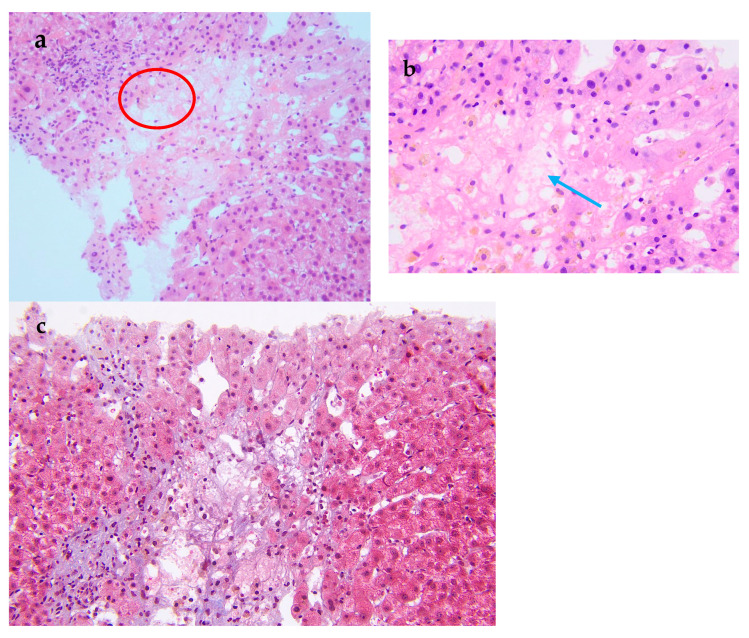
Pathological examination of the liver biopsy. (**a**) The central vein injury is accompanied by bleeding and occasionally intravascular micro-thrombosis (red circle). (**b**) Kupffer cells phagocytosing pigment granules are visible within the hepatic sinusoids (blue arrow). (**c**) Masson trichrome stain demonstrates mild periportal fibrosis and occasional enlarged portal tracts, findings diagnostic of acute HSOS. Relevant histopathological features consist of heterogeneous liver parenchyma, attenuated hepatic veins, and multiple liver nodules. A 19-year-old man was admitted to the hospital with 3 months of malaise, poor appetite, and nausea, as well as abdominal distension, scanty urination, and a yellowish color, and these symptoms gradually worsened. Laboratory tests at the local hospital suggested elevated liver enzymes, and an ultrasound showed hepatomegaly and ascites. Since then, the patient has experienced intermittent fevers at night, and the temperature returned to normal in the morning. He was given hepatoprotection, peritoneal puncture, and hemodialysis. After that, his creatinine was gradually reduced to normal, and the liver function improved significantly. However, the patient had a fever again and was admitted to our hospital after 1 month. For past medical history, he had a nine-year history of aplastic anemia (AA). He was treated with anti-thymocyte globulin, cyclosporine A, and androgen. The patient has been undergoing long-term follow-up treatment in the hematology department. The patient’s hematological condition flared up as soon as he stopped taking his medication, so he took regular oral cyclosporine 1000 mg in the morning and 750 mg in the evening, and androgen tablets 80 mg in the morning and 40 mg in the evening. After admission to our hospital, his physical examination revealed a malnourished appearance, a yellowish skin tone, and a distended abdomen with ascites, giving a frog-like appearance. Shifting dullness could be elicited. His temperature was 37.4 °C; heart rate, 120/min; respiratory rate, 18/min; blood pressure, 138/87 mmHg. Laboratory tests showed significantly elevated alanine aminotransferase, 2918 U/L (reference range, 9–50 U/L [0.15–0.8 μkat/L]); alkaline phosphatase, 143 U/L (reference range, 15–40 U/L [0.25–0.67 μkat/L]); lactate dehydrogenase, 2212 U/L (reference range, 109–245 U/L [7.27–4.08 μkat/L]); total bilirubin 103 μmol/L (reference range, 3.0–21.0 μmol/L); direct bilirubin 71 μmol/L (reference range, 0.0–7.0 μmol/L); and serum creatinine, 520 μmol/L (reference range, 59–104 μmol/L). Hemoglobin level was 6.7 g/dL (reference range, 13.0–17.5 g/dL); haptoglobin < 5.83 mg/dL (reference range, 36.00–195.00 mg/dL), plasma free hemoglobin level was 33 mg/L (reference range, 0–40 mg/L); D-dimer, 32,607 ng/mL (reference range, 0–243 ng/mL), and Coombs test was negative. Ascite tests indicated transudate. He underwent peripheral blood flow cytometry, which showed an extremely high proportion of paroxysmal nocturnal hemoglobinuria (PNH) clones, with type III cells (completely missing CD59) at 94.3%, and type II cells (partially missing CD59) at 1.4%. He was diagnosed with aplastic anemia/paroxysmal nocturnal hemoglobinuria (AA/PNH) syndrome. An abdominal computed tomography scan with intravenous contrast revealed that the density of the enlarged liver was homogeneous, with a low-density region that was enhanced lower in the arterial phase. Additionally, the three hepatic veins were not clearly visible, and a large volume of ascites was present. A transcutaneous liver biopsy showed central vein injury, and bleeding was commonly seen in the combination of intravascular micro-thrombosis and hepatic sinusoid dilatation was partially found. Computed tomography (CT) and the liver biopsy led to a final diagnosis of hepatic sinusoidal obstruction syndrome (HSOS) to AA/PNH syndrome. The therapeutic strategies for PNH are complement inhibition therapy (eculizumab) and bone marrow transplantation. Eculizumab, a monoclonal antibody that blocks terminal complement at C5, has changed the natural history of PNH. However, as it was approved in 2022 in China, this patient did not receive Eculizumab therapy. He was treated with diuretics and intermittent abdominal paracentesis. He was also initiated on subcutaneous injections of low-molecular-weight heparin 4000 IU every 12 h, followed by transition to oral warfarin 3 mg per day for long-term anticoagulation therapy. Methylprednisolone 40 mg per day was used for PNH and then transitioned to oral prednisolone 50 mg per day. The dose of corticosteroids was decreased to 40 mg per day after the hemoglobin was stabilized. Eventually, the patient’s ascites and jaundice disappeared, and his liver function improved. Discussion: Thrombosis is the most frequent clinical complication and also the main cause of mortality in PNH [1]. The incidence of thrombus in PNH was 10.61/100 patient-years. There was a strong correlation between thrombotic events and a high proportion of PNH clones, with a 10 percent increase in PNH clones associated with a 1.64-fold increase in the risk of clots [2]. In this case, the peripheral blood flow cytometry showed an extremely high proportion of PNH clones, with type III cells (completely missing CD59) at 94.3% and type II cells (partially missing CD59) at 1.4%. According to the established criteria, a granulocyte clone size of this magnitude (>50%) is diagnostic of classical PNH and is unequivocally associated with a high risk of life-threatening thrombotic events [3]. Therefore, this laboratory result is consistent with a diagnosis of high-risk classical PNH. HSOS is characterized by particular, non-thrombotic, microvascular lesions, which may extend to the central vein in the liver. Ingestion of certain toxic alkaloids, high-dose radiotherapy or chemotherapy, and organ transplantation are all risk factors for HSOS. Clinical manifestations include quadrant abdominal pain, ascites, hepatomegaly, increased weight gain, and jaundice. Imaging can reveal hepatomegaly, ascites, slowed or reversed hepatic/portal blood flow, and widening of the portal vein [4]. Liver biopsy is the gold standard for the diagnosis of HSOS and is recommended when clinical and imaging findings are not sufficient to make a diagnosis. A transcutaneous liver biopsy showed central vein injury, and hemorrhage was commonly seen in the combination of intravascular micro-thrombosis and hepatic sinusoid dilatation was partially found, which indicated HSOS in this case. However, it must be emphasized that PNH leading to HSOS was particularly rare. The rarity lies in the distinct pathogenesis of microvascular lesions: systemic complement-mediated thrombosis in PNH versus direct toxic injury to sinusoidal endothelial cells in HSOS. First, in PNH patients, thromboses tend to originate in the venous system, especially in the hepatic and cerebral venous vessels [5], unlike the hepatic microvascular occlusion of HSOS. Second, the initiating mechanisms are fundamentally different. The earliest morphological change in HSOS is sinusoidal endothelial cell injury induced by glutathione depletion, nitric oxide depletion, increased matrix metalloproteinases, increased vascular endothelial growth factor, and possibly clotting factors (Appendix A). In contrast, the mechanisms of thrombophilia in PNH included complement-mediated intravascular hemolysis, impaired nitric oxide bioavailability, activation of PNH platelets, and impairment of the fibrinolytic system, which consequently formed fibrin-platelet microthrombi [1]. Conclusion: We have reported a rare case of AA/PNH that presented as hepatic sinusoidal obstruction syndrome and resolved following effective anticoagulation and corticosteroid treatment. This case underscores the critical need for awareness of HSOS secondary to AA/PNH syndrome, which requires a high index of suspicion and for which prompt treatment is needed to reduce morbidity and mortality.

## Data Availability

Data will be made available from the corresponding author upon request.

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
