# Peer review of "Hepatic Sinusoidal Obstruction Syndrome Secondary to Aplastic Anemia/Paroxysmal Nocturnal Hemoglobinuria Syndrome: A Rare Case"

_diagnostics, 2025, doi:10.3390/diagnostics15212712_

Round 1
Reviewer 1 Report
Comments and Suggestions for Authors
This manuscript presents a rare and clinically significant case of hepatic sinusoidal obstruction syndrome (HSOS) secondary to aplastic anemia/paroxysmal nocturnal hemoglobinuria (AA/PNH) syndrome. The report is well-structured and highlights important diagnostic, clinical, and therapeutic aspects. The case provides value to the readership of Diagnostics as it underscores the diagnostic challenges and therapeutic considerations in rare hematological-liver disease intersections.
However, certain areas require further clarification, expansion, and stylistic improvements to strengthen the manuscript’s scientific rigor and readability.
Major Comments
- The manuscript addresses a very rare complication (HSOS secondary to AA/PNH syndrome). This novelty is commendable and makes the case worth publication. However, the discussion could be expanded to place this case in the context of previously reported similar cases (if any). A literature comparison highlighting incidence, outcomes, and therapeutic strategies would increase the scientific depth.
- The clinical presentation is described in detail, but the timeline could be better structured. For instance, initial symptoms, progression, investigations, and interventions should be presented in a clearer chronological sequence to improve readability.
- Abbreviations such as HSOS, AA, and PNH should be consistently defined at first mention in the main text, not just in the abstract.
- The manuscript mentions peripheral blood flow cytometry, CT, and liver biopsy as diagnostic tools. The flow cytometry results are impressive (94.3% type III clones), but the clinical significance of such a high burden should be elaborated in the discussion with references.
- The authors should provide more detail on the histological findings of the liver biopsy (e.g., images with labeling, and a brief pathological description beyond central vein injury and micro-thrombosis).
- The management strategy is described (anticoagulation, corticosteroids, supportive care). However, the rationale for the choice of corticosteroids and the tapering scheme could be better explained.
- Was eculizumab considered or not available? Given that it is the mainstay therapy for PNH in many countries, a brief explanation is important.
- The discussion appropriately highlights thrombosis as the major complication of PNH, but it could benefit from a more focused analysis of why microvascular hepatic thrombosis leading to HSOS is particularly rare.
- A table contrasting common thrombotic sites in PNH versus rare manifestations (like HSOS) could improve clarity.
Minor Comments
- The abstract is concise but slightly repetitive (mentions thrombosis as the most frequent complication multiple times). Condensation and sharper emphasis on the rarity of HSOS would improve impact.
- Figure legends are too brief. More descriptive legends with clinical correlation are needed (e.g., indicate arrows or markers on CT images and biopsy images).
- There are grammatical inconsistencies (e.g., “thromboses are more tend to originate” → should be “thromboses tend to originate”). A thorough language edit is recommended.
- Replace informal descriptions (e.g., “frog-like abdomen”) with more clinically appropriate terminology (“distended abdomen with ascites”).
- The reference list is adequate but limited to four key papers. Adding more recent literature on HSOS in hematological disorders would strengthen the background and discussion.
- The informed consent statement is clear. Please ensure that the Institutional Review Board (IRB) or ethics approval status is explicitly mentioned (even if waived for case reports).
Recommendation
Major Revision
This is an interesting and rare case with educational value. However, before acceptance, the authors should:
- Expand the discussion with literature context and clarify the therapeutic rationale.
- Improve figure quality and legends.
- Edit for grammar and style.
- Provide more detail on the liver biopsy findings and consider adding histopathology images with annotations.
Reviewer 2 Report
Comments and Suggestions for Authors
There are some comments.
It would be beneficial to describe the clinical significance and compare it with prior literature in more detail.
It would be beneficial to describe the histopathological features in more detail.
Regarding Figure 2, the H&E-stained photo does not align well with the figure legend. Intravascular microthrombosis is not identified. Please check or add histopathological images.
Please check reference formatting according to the Journal diagnostics.
Please check English grammar and spelling.
For example, thromboses are more tend to originate -> thromboses tend to originate
ng/ml -> ng/mL.
40mg/day -> 40 mg/day
3mg per day -> 3 mg per day”.
Round 2
Reviewer 1 Report
Comments and Suggestions for Authors
Thank you for addressing all the comments. I would like to accept the revised version of the manuscript
Author Response
We wish to express our sincere gratitude for your time and for the final positive decision on our manuscript. We are delighted and honored that you found our work acceptable for publication.
Your insightful comments and constructive suggestions throughout the review process have been invaluable in helping us significantly improve the quality of our paper.
Thank you again for your thorough and thoughtful review.
Reviewer 2 Report
Comments and Suggestions for Authors
There are two main points to note.
1. The histopathological features were not described sufficiently. It would be better to describe these features in more detail.
2. In Fig. 2, the microthrombosis (red circle) and the Kupffer cells phagocytosing pigment granules (blue arrows) are not clear. It would be better to replace these with clearer photos.
Comments on the Quality of English LanguagePlease check English grammar and spelling.
For example, in Table 1, ... HSOH -> HSOS
Round 3
Reviewer 2 Report
Comments and Suggestions for Authors
The manuscript has been thoroughly revised.
However, Figure 2 is still not suitable for publication.
It would be better to replace it with more explicit photos.
